# Mobile Application for Visible Light Communication Systems: An Approach for Indoor Positioning

**Quan Dinh Nguyen** [1] and **Nam Hoang Nguyen** [2,*]

1   School of Electrical and Electronic Engineering, Hanoi University of Science and Technology (HUST), Hanoi 100000, Vietnam; nguyendinhquan140701@gmail.com
2   Department of Automation, School of Electrical and Electronic Engineering, Hanoi University of Science and Technology (HUST), Hanoi 100000, Vietnam
*   Correspondence: nam.nguyenhoang@hust.edu.vn

**Abstract:** We explore the use of smartphones to decode data transmitted from LEDs to smartphone cameras in visible light communication (VLC) applied to indoor positioning applications. The LEDs—modified to enable rapid on-off keying—transmit identification codes or optically encoded location data imperceptible to human perception. Equipped with a camera, the smartphone employs a single framed image to detect the presence of the luminaires in the image, decode their transmitted identifiers or locations, and determine the smartphone's position and orientation relative to the luminaires. The camera captures and processes images continuously. The following fundamental issues are addressed in this research: (i) analyzing the camera parameters on smartphones that affect data decoding results; (ii) exploiting the rolling shutter effect of the CMOS image sensor to receive multiple bits of data encoded in the optical communication line with a single frame shot; (iii) advancing research in developing algorithms to process data from multiple LEDs simultaneously. We conduct experiments to evaluate and analyze feasibility, as well as the challenges of the design, through scenarios varying in distance, transmission frequency, and data length.

**Keywords:** visible light communication (VLC); on-off keying; indoor positioning; CMOS image sensor; image processing

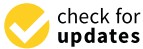

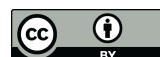

## 1. Introduction

Historically, wireless communication technology, predominantly based on radio frequency (RF), has been the cornerstone of various wireless applications. However, the exponential rise in the popularity of smart mobile devices has begun to strain the limits of RF spectrum usage currently [1]. Optical wireless communication (OWC) has been considered to reduce bandwidth usage pressure on RF wireless systems in a harmonious manner when bandwidth is not limited. Optical wireless communication (OWC) systems will be part of future heterogeneous networks as a complementary option to RF and are competing to be included in 5G/6G networks. OWC, which encompasses ultraviolet (UV), visible light (VL), and infrared (IR) spectrums, offers a wide theoretical bandwidth ranging from 300 GHz to 430 THz (IR), 430 THz to 790 THz (VL), and 30 PHz (UV), as shown in Figure 1 [1–3]. This bandwidth significantly exceeds the limited range from 3 KHz to 300 MHz in RF (radio frequency) technology. OWC includes diverse transmission methods suitable for indoor and outdoor environments. These methods consist of infrared communications (IRCs), Free Space Optics (FSOs) for communication in open space, and visible light communication (VLC), as shown in Figure 2 [1,4,5].

During its nascent stages, OWC mainly focused on applications in military and deep space communications, with relatively limited use in civilian sectors [6,7]. However, the landscape of OWC has experienced a noteworthy shift in recent years, particularly with the emergence of diffuse infrared communications, which presented an appealing solution for short-range data exchange [8,9]. Initially, this technology found its niche in devices like

TV remote controls. Nevertheless, the field has undergone a remarkable transformation with the development of visible light communications (VLCs), made possible by the development of white light LEDs and the available lighting infrastructure. VLC's dual role as both a means of illumination and data communication has broadened the horizons of OWC, unveiling substantial potential and opportunities for innovation. Light from the LED is modulated for data transmission, which is then detected using a photodiode (PD) or image sensor (IS). VLC's versatility has paved the way for the creation of intelligent lighting systems, the establishment of vehicle-to-vehicle networks, and the provision of location-based services, offering promising prospects for both indoor and outdoor applications. This evolution in OWC technology is poised to drive further advancements and applications in the realm of optical data transmission.

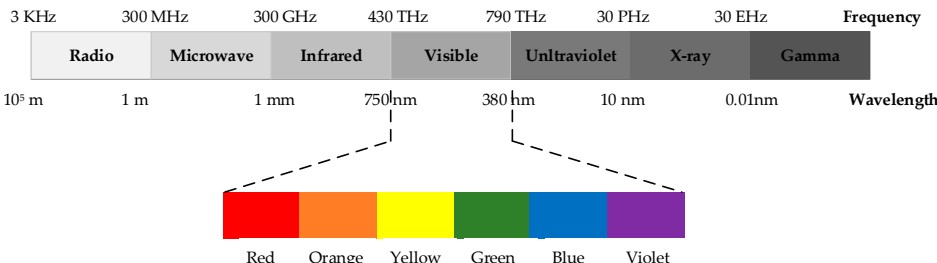

**Figure 1.** The electromagnetic spectrum.

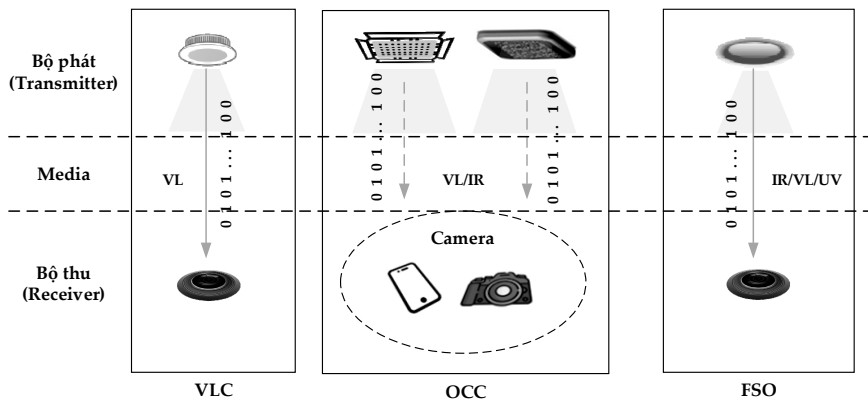

**Figure 2.** Optical wireless communication (OWC) technologies.

## 1.1. Recent Development of Visible Light Communication

Visible light communication (VLC) has emerged as a revolutionary technology that utilizes the rapid modulation capabilities of light-emitting diodes (LEDs) for high-speed data transmission as well as illumination purposes. The origins of VLC can be traced back to 1999 [10], marking its early development and subsequent advancements over the years. In 2011, the IEEE standard for VLC, designated as IEEE 802.15.7 [11], was published, providing essential guidelines for industrial VLC applications, further propelling the field [12]. Li-Fi was known in 2013 and has facilitated collaborative research with industry partners, emphasizing the industry's growing interest in VLC technology [13].

Visible light communication (VLC) is a promising wireless communication method with advantages over traditional radio frequency (RF) communication. While the RF spectrum is primarily allocated for civil and military wireless communication, VLC enjoys the absence of spectrum regulations, making it a versatile and interference-free option [14–16]. It utilizes efficient and long-lasting light-emitting diodes (LEDs), commonly used for lighting and indication. LEDs, known for their high efficiency and long lifespan, have emerged as an ideal choice for widespread lighting and indication applications, further enhancing the appeal of VLC systems. The seamless integration of VLC with existing

LED-based lighting infrastructure, owing to the ease of LED modulation, makes it a cost-effective and efficient choice for both indoor and outdoor smart lighting systems, as well as location-based services [12]. Beyond lighting applications, VLC demonstrates versatility by extending its reach into realms such as robotics, industrial applications, and its role as a wireless connectivity solution for the Internet of Things (IoTs), reflecting its potential to shape the future of communication technology [10,17–19].

### 1.2. Comparison between Visible Light Communication and Radio Frequency

Given their basic characteristics, there are notable differences between visible light communication (VLC) and radio frequency (RF) systems. The advantages and disadvantages of these two technologies are presented as follows:

- The distinctive nature of visible light communication (VLC) as a communication medium, characterized by noncoherent LED light sources, presents a significant difference from the traditional modulation techniques utilized in radio frequency (RF) communications. In RF systems, modulation encompasses the manipulation of amplitude, frequency, or phase to convey information, offering a wide range of possibilities. In contrast, VLC mainly relies on intensity modulation and direct detection as the primary method for data transmission. This approach imposes specific constraints, restricting the transmission of only real, non-negative signals in VLC, aligning with the unique characteristics of light as the medium for communication. This divergence underscores the need for innovative signal processing techniques and communication strategies tailored to VLC, opening new orientations for research and development in the field of optical wireless communication.

- Indoor visible light communication (VLC) systems are often seamlessly integrated into existing lighting infrastructure, making illumination a fundamental and integrated aspect of VLC technology. Within a typical office environment, the recommended illumination level typically falls around 400 lux [20]. In such settings, a VLC system is expected to provide stable and non-flickering lighting that can be dimmed for control and to meet the desired ambiance. In VLC, the optical power transmitted serves a dual purpose, as it is not solely dedicated to communication but also fulfills the crucial role of illumination. Therefore, the design criterion for VLC systems shifts away from the conventional RF systems, where the emphasis is primarily on minimizing the transmitted power. The development of VLC technology in this context not only enhances communication capabilities, but also contributes to energy-efficient and aesthetically pleasing indoor lighting solutions.

- In contrast to radio waves, which have the ability to penetrate walls and obstacles, optical signals, including those employed in visible light communication (VLC), are unable to traverse physical barriers. This inherent characteristic of optical signals brings a significant advantage in terms of security, as these signals tend to remain confined within the physical boundaries of the space in which they are emitted. When considering long-distance VLC communications, the concept of line-of-sight (LoS) emerges as an essential requirement. In LoS scenarios, both the sender and receiver must maintain an unobstructed, direct line-of-sight to each other for effective communication. Any intervening obstruction or barrier in the transmission path becomes immediately evident and can disrupt the communication link. Consequently, VLC technology garners substantial favor in military and governmental applications where stringent information privacy and security are of paramount importance. In these contexts, VLC's ability to preserve the integrity of communication within confined spaces, comes with its susceptibility to obstructions, rendering it a highly desirable choice for the protection of sensitive information and the establishment of secure communication channels.

- The challenge of indoor localization has become a prominent and actively researched area, drawing significant attention in recent years. The use of global navigation satellite system (GNSS) signals for indoor localization faces practical challenges, primarily

stemming from the effects of multipath transmission and signal attenuation within indoor environments. In the field of indoor positioning, conventional approaches have traditionally centered around radio-frequency methods, sound signals, and camera-based systems [21]. However, recent research has unveiled that visible light communication (VLC) systems can provide surprisingly accurate solutions for indoor localization applications [22]. VLC-based positioning (VLP) systems have proven to be highly effective in settings such as hospitals, airports, museums, and industrial plants, where traditional technologies often encounter limitations and constraints. This innovative application of VLC technology not only enhances indoor localization accuracy, but also opens new possibilities for precise tracking and navigation in challenging indoor environments.

*1.3. Paper Contribution*

In the previous presentations, we discussed some background research on the VLC system: the development of VLC in recent years, the applications of VLC, and the advantages and disadvantages of VLC compared to RF. This research paper aims to discuss different approaches to indoor positioning technologies and propose an indoor positioning system that applies the advantages of VLC technology. In the remaining parts of the article, we will present the following structure: Section 2 presents the architectural overview of a VLC system including the transmitter, receiver, and layer architecture. In Section 3, the literature review focuses on previous research on modulation techniques for VLC systems and technologies for indoor positioning such as Wi-Fi localization, radio-frequency identification, Bluetooth, and VLC systems. Section 4 presents the design of a VLC system applied to indoor positioning, followed by detailed experimental results discussed in Section 5. Finally, Section 6 presents conclusions based on the acquired results and the potential future development directions.

## 2. Visible Light Communication System Architecture

The VLC system consists of two main components: the transmitter and the receiver, often comprising three common layers. These are the physical layer, the MAC layer, and the application layer. The architecture of the VLC layers is illustrated in Figure 3 [23]. In the IEEE 802.15.7 standard, only two layers (PHY and MAC) are defined for simplicity in categorization [24].

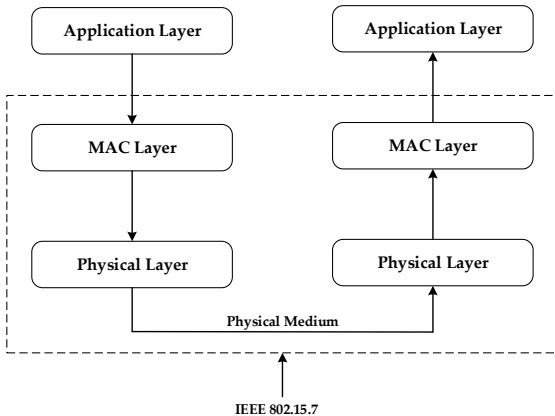

**Figure 3.** Layered architecture of VLC.

*2.1. MAC and Physical Layer*

2.1.1. MAC Layer

The topologies supported by the MAC layer include point-to-point, broadcast, and star [25]. In a point-to-point topology, there are only two participating nodes, and dedicated links are established to build a communication network. Concerning a broadcast topology, there are multiple participating nodes within a single link, but a single node transmits

messages to all other nodes simultaneously. A star topology is a topology with one central station more important than all other nodes, controlling the communication activities of the entire network. It can be described as a combination of the point-to-point and one-to-many network architectures, as shown in Figure 4.

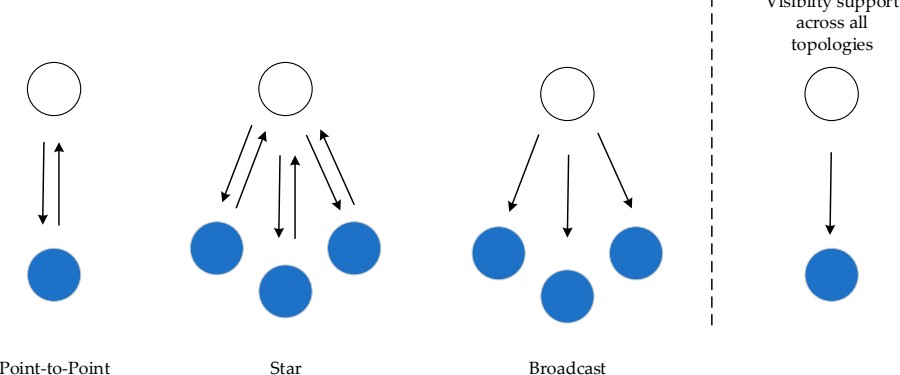

**Figure 4.** The topologies of MAC layer.

2.1.2. Physical Layer

The physical layer defines the device's physical specifications and its interaction with the medium, as shown in Figure 5.

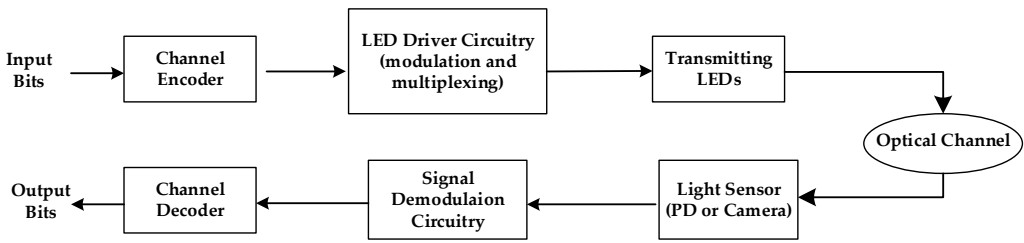

**Figure 5.** Physical layer system model of VLC.

Initially, the input bits may pass through optional encoding before proceeding to the modulation stage, which includes techniques like on-off keying, PPM, and PWM. Then, the modulated data are transmitted using an LED through the optical channel [26].

*2.2. Transmitter*

Among popular light sources such as fluorescent lamps, incandescent lamps, and halogen lamps, only LED lamps are currently deemed suitable for transmitting light in space using VLC technology. In comparison to traditional light sources, LEDs outshine them in terms of reliability, longevity, and luminous efficiency. It offers a broader color spectrum and flexible control capabilities. The efficiency of LED lamps is evident in their substantial advantage of over 20 Lm/W when compared to traditional incandescent lamps [27]. Additionally, the LED's rapid switching capability between on and off states is a significant advantage, particularly in VLC applications and positioning tasks.

LEDs and lasers serve as transmission sources for VLC. LEDs are preferred when a single device needs to handle both communication and illumination. Presently, white light generated by LEDs stands out as an attractive choice for VLC sources. The popular technique utilized to produce white light using LEDs involves trichromatic methods, typically using red, green, and blue (RGB) components. Utilizing an RGB LED for white light production offers the benefit of high bandwidth and, consequently, elevated data transmission rates. However, this approach comes with increased complexity and modulation challenges. Table 1 presents a comparison between phosphor-based LEDs and RGB LEDs.

In prior works [8,28,29], various methodologies were employed to characterize the optical wireless channel, and LED selection was based on the specific channel model.

**Table 1.** Comparison between phosphor-based LEDs and RGB LEDs.

|  | RGB LEDs | Phosphor-Based LEDs |
|---|---|---|
| Data rates | Up to 100 Mbps | Up to 50 Mbps |
| Price | More expensive | Less expensive |
| Modulation | Complex | Low complexity |
| Bandwidth | High | Low |

*2.3. Receiver*

In current VLC systems, one of two receiver types is used: a photodiode (PD) or an imaging sensor. Generally, photodiodes are the standard choice for capturing VLC signals and are typically used in fixed receiver setups. However, for mobile scenarios with wider fields of view (FOV), image sensors are preferred over photodiodes. Thus, this paper will focus on an image sensor-based VLC system implemented on smartphones, as the primary objective is to deploy them in indoor positioning applications involving continuous mobility.

The image sensor (IS) in a smartphone camera serves as a critical component responsible for detecting and converting light signals into electronic data. This capability enables the camera to capture and process images of the subjects [30]. In a camera, the shutter mechanism determines the exposure of pixels on the IS. There are two primary IS architectures: CCD and CMOS. CCD sensors have a larger physical size due to the use of a relatively larger analog-to-digital converter (ADC) compared to CMOS IS [31,32]. Consequently, CCD sensors are not commonly used in mobile phones, especially smartphones. In contrast, CMOS cameras offer advantages such as lower power consumption, reduced IS size, faster readout, lower cost, and greater programmability. Additionally, in CMOS architecture, pixel scanning is based on the X-Y addressing scheme, allowing direct access to specific IS components. The cameras utilize two distinct exposure methods: global shutter (GS) and rolling shutter (RS), as shown in Figure 6. In global shutter-based IS, the entire sensor is exposed to light simultaneously, while a rolling shutter-based camera exposes each row to light sequentially, resulting in a reduction in overall brightness.

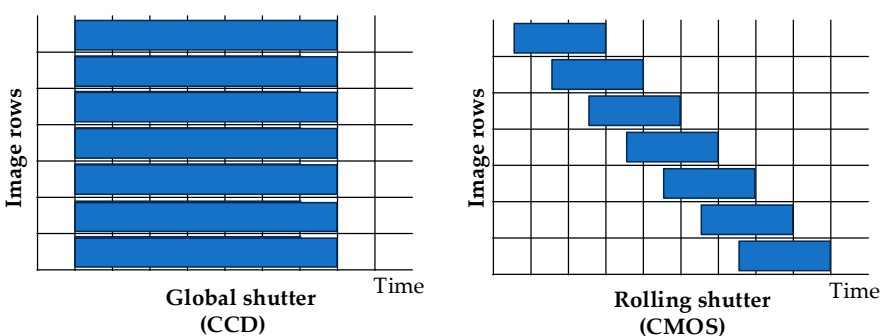

**Figure 6.** Difference between the CCD and CMOS.

Most CMOS image sensors embedded in mobile phones are operated using a rolling shutter. This means that each row of pixels in the CMOS sensor is activated sequentially, and it does not capture the entire image at once (global shutter), as shown in Figure 6. Although this possibly has an undesirable effect when recording video of moving subjects, it is beneficial to the VLC system because the data rate can be much faster than the frame rate (FPS). Therefore, an image can represent many logical bits.

## 3. Literature Review

In this section, we focus on discussing previous studies on modulation techniques in VLC systems and comparing indoor positioning technology using VLC systems with RF-based indoor positioning systems. The research on indoor positioning based on radio frequency is presented as follows: Wi-Fi positioning is discussed, followed by research on radio-frequency identification and Bluetooth positioning. Then, studies on the VLC system using different position estimation techniques are discussed. Finally, a discussion and comparison of the technologies are presented.

### 3.1. Modulation Techniques for Visible Light Communication

In the physical layer, coding and modulation algorithms have been developed to support high-speed and reliable optical wireless connections. Intensity modulation, such as on-off keying (OOK) LEDs, are used in VLC systems to convey data by toggling between an 'on' state and an 'off' state, where the 'on' state represents a binary '1', and the 'off' state represents a binary '0'. In on-off keying (OOK), the LED is not completely extinguished in the 'off' state but instead undergoes a reduction in intensity. OOK is favored for its straightforward implementation.

Previous VLC research has utilized white LEDs, combining blue emitters with yellow phosphors. However, this approach faced limitations related to bandwidth due to the relatively slow response time of the yellow phosphor [33]. Notably, a data rate of 10 Mbps was achieved using Non-Return-to-Zero (NRZ) OOK with a white LED in prior research [34]. To overcome bandwidth limitations, researchers explored the combination of analog equalization with blue filtering, which significantly enhanced data rates to 125 Mbps and 100 Mbps, respectively [35,36].

The limitation of on-off keying (OOK) lies in its relatively low data rates, which has spurred researchers to explore novel modulation techniques capable of achieving higher data rates. One such approach is Pulse Width Modulation (PWM), where the width of pulses varies in accordance with dimming levels. By utilizing a high PWM frequency, a range of dimming levels can be achieved spanning from 0% to 100% [37]. However, the PWM method presented in [38] had its limitation, capping the data rate at 4.8 Kbps. In response to these limitations, researchers in [39] combined PWM with Discrete Multitone (DMT) to enable both communication and dimming control, resulting in a higher data rate than that of [37]. Another modulation scheme, Pulse Position Modulation (PPM), is based on the position of the pulse within the symbol duration. PPM divides the symbol duration into equal time intervals or slots (t slots) and transmits the pulse within any of these slots. Nonetheless, PPM suffers from the constraint of low data rates since only a single pulse can be transmitted in each symbol period. To enhance spectral efficiency, Multi-Pulse PPM (MPPM) was introduced, which allows for the transmission of multiple pulses within each symbol-time, making PPM more spectrally efficient.

In addition, Color Shift Keying (CSK) was introduced as part of the IEEE 802.15.7 standard with the aim of enhancing data rates, particularly addressing the limitations found in other modulation schemes [25]. CSK achieves modulation by manipulating the intensity of light emitted from LEDs. Traditionally, this modulation technique was hampered by the delay in switching due to the generation of white light using a combination of yellow phosphor and blue LEDs. An alternative approach to producing white light involves utilizing three distinct LEDs emitting green, blue, and red light. In CSK, modulation is accomplished by varying the intensity of these three colors within an RGB LED source. CSK relies on the color space chromaticity diagram, which maps all colors that can be perceived by the human eye to two chromaticity parameters, typically denoted as 'x' and 'y'.

### 3.2. Technology for Indoor Positioning

3.2.1. Indoor Positioning Systems Utilizing Radio-Frequency Technology

Wi-Fi stands as one of the most widely utilized technologies for indoor localization, often paired with techniques reliant on received signal strength or fingerprinting. In [39],

Kunhoth explored the realm of Wi-Fi-based indoor navigation systems, spurred by the challenges posed by the fluctuations in received signal strength (RSS) that affected positioning accuracy. To address this issue, Kunhoth introduced the concept of a fingerprint spatial gradient (FSG) as a solution. It is noteworthy that over the past decade, many machine learning algorithms, including Support Vector Machine (SVM), k-Nearest Neighbor (KNN), and neural networks, have found application in pattern matching for indoor localization methods based on radio fingerprints [39]. Another technology frequently used for indoor localization is radio-frequency identification (RFID). Bouet et al. delved into the utilization of RFID technology as one of the communication-centric methods for indoor positioning [40]. An RFID system requires both an RFID tag and an RFID reader to function. Notably, for indoor RFID applications, passive tags are often used, eliminating the need for an external power source. RFID technology employs range-based techniques such as Angle of Arrival (AOA), RSS, Time of Arrival (TOA), and Time Difference of Arrival (TDOA) to estimate positions within indoor settings. Among these, RSS stands out for its ability to provide position estimations even in non-line-of-sight scenarios, making it a valuable approach for indoor positioning systems [40].

Bluetooth technology has undergone a significant transformation in the realm of indoor localization methods, with extensive research dedicated to its application. In a survey conducted by Kunhoth [39], it was highlighted that Bluetooth Low-Energy beacons serve as radio-frequency signal sources to aid in user positioning, offering accuracy levels comparable to Wi-Fi-based systems.

### 3.2.2. Indoor Positioning based on the Visible Light Communication System

Recently, visible light communication (VLC) has emerged as a pivotal technology, finding critical applications in physical space navigation and object identification within constrained environments, both at individual and industrial scales. VLC offers a means of achieving precise indoor positioning with a relatively uncomplicated system setup. In this context, this study examines research efforts that have worked with both modified and unmodified light sources to assess the respective advantages and drawbacks of each approach.

For instance, in the work presented by Zhang et al. in [41], a VLC-based indoor localization system was introduced, designed to account for sunlight exposure factors. The system employs a triangulation method based on received signal strength (RSS). Notably, the transmitters in this setup are modified LEDs that operate without the need for synchronization, eliminating the requirement for interconnection between transmitters. As a result, this system offers a straightforward and cost-effective deployment solution for indoor environments. However, it is worth noting that the system employs multiple transmitters and a single receiver, introducing a channel multi-access challenge. This challenge is addressed by utilizing an asynchronous channel multiplexing technique known as basic framed slotted ALOHA (BFSA) to detect the light sources. Despite this complexity, the system demonstrated promising outcomes, achieving a precision of 95% within 17.25 cm under direct sunlight exposure and a precision of 95% within 11.2 cm under indirect sunlight exposure.

Li et al. introduced a visible light localization system in their study documented in [42]. This system comprises two primary components: an LED bulb and a receiving device, typically a smartphone. These components house functional modules designed to accomplish three essential technical aspects: light beaconing, distance estimation, and localization. The light beaconing component functions by employing specially modified LED bulbs to transmit location information to the receiver. To achieve this, the LEDs use a binary frequency shift keying (BFSK) modulation module to encode messages, which are subsequently demodulated at the receiver's end. Simultaneously, the distance estimation component on the receiver side deciphers the information conveyed by the light beacons emitted from multiple light sources, measuring their received signal strength (RSS). The localization component then leverages trilateration or multilateration, depending on the

number of visible light sources, to pinpoint the receiver's exact location. The findings in this paper underscore the substantial potential of using visible light for achieving high-precision indoor localization.

In [43], Zhao et al. introduced a visible light localization system named 'NaviLight', which innovatively harnesses unmodified light sources for the purpose. This system capitalizes on pre-existing lighting fixtures as transmitters and adopts light intensity as a unique identifier, referred to as the 'LightPrint', to ascertain a user's precise location. NaviLight's concept draws inspiration from Wi-Fi-based indoor localization systems that rely on the received signal strength indicator (RSSI) fingerprint to pinpoint user locations. Nevertheless, utilizing light intensity as a fingerprint in a VLC system presents unique challenges. In contrast to electronic signal strength, light intensity is inherently more coarse-grained and less precise across spatial dimensions. Furthermore, in contrast to Wi-Fi systems, there is no direct communication between the light source and the receiver. To surmount these challenges, NaviLight incorporates a vector of multiple light intensity values in conjunction with user movement patterns to determine location. Leveraging existing lighting infrastructure, this system is renowned for its ease of deployment, cost-effectiveness, and high accuracy. It adapts seamlessly to various indoor environments. However, a potential limitation is that light intensity information can be insufficient, as it may not adequately differentiate between locations with similar light intensity. Additionally, the computational cost of matching LightPrints to pre-trained data for position determination can be significant.

In their work documented in [44], Kuo et al. introduced a novel positioning method named 'Luxapose', which utilizes LED lights in conjunction with smartphones for accurate location determination. In this method, the LEDs are modified to emit optical pulses containing unique location information that remains imperceptible to the human eye. Utilizing an unmodified smartphone camera, images are captured to detect the presence of the light source, decode the embedded identifiers and positions, and subsequently estimate the smartphone's precise location and direction in relation to the LED lamp. The experimental outcomes of this approach revealed its potential for achieving localization errors at the decimeter level, alongside a 3-degree orientation error, particularly when users move beneath overhead LED lights.

### 3.2.3. Discussion

Indoor localization technology is a prominent research topic spanning many different areas, with many practical applications. Many solutions have been put forward for indoor localization, including aids for the visually impaired and navigation, wayfinding systems in indoor spaces, such as shopping malls, museums, and airports, and geolocation asset location in corporate and warehouse environments [39].

However, Rahman et al. [45] identified some important challenges associated with radio frequency-based indoor positioning technology. These include significant power consumption, security concerns, and limited throughput. RFID and Bluetooth, when utilized as indoor positioning techniques, exhibit low power consumption characteristics. Nonetheless, it is essential to note that Bluetooth's coverage range is relatively limited, whereas RFID has its own set of constraints, including the necessity for an RFID reader, extended response times, and limitations in terms of user adaptability.

In contrast, Wi-Fi, while offering certain advantages, comes with a high-power consumption factor and considerable deployment costs when fingerprinting methods are employed for building databases. On the other hand, VLC-based positioning systems offer a series of compelling advantages. They can be cost-effectively installed, primarily relying on existing lighting infrastructure with minimal modifications required. Consequently, the implementation of VLC significantly enhances the ability to identify objects within physical spaces.

## 4. Visible Light Communication System Design for Indoor Positioning

Our goal is to design a VLC system for indoor positioning where the LED lighting infrastructure shown in Figure 7 is used [6]. This approach is based on assigning a unique identification code (ID) to each LED cluster and using smart devices, such as smartphones with cameras using CMOS sensors, to decode the IDs. From there, the system will query the received ID on the database to locate the standing position and give directions to the desired location.

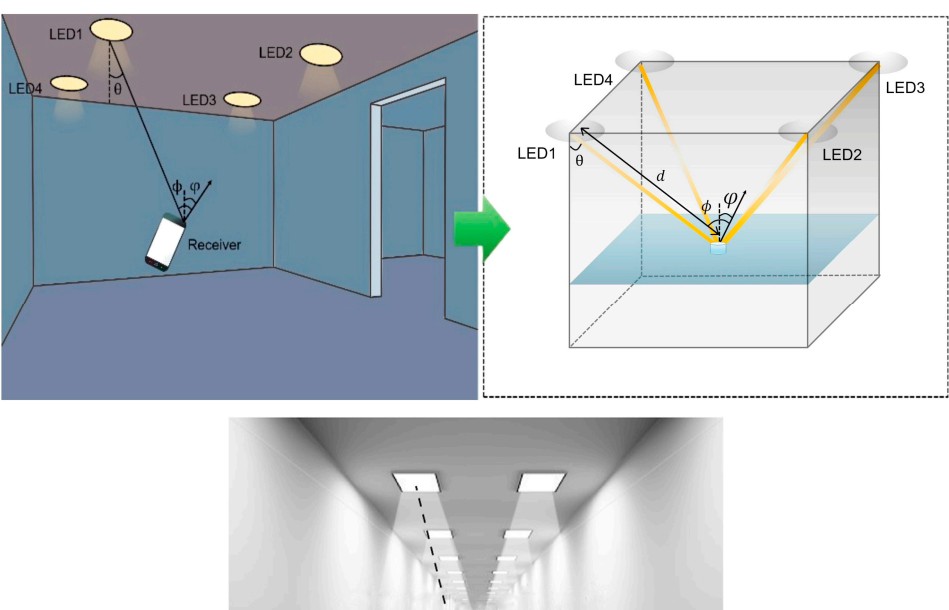

**Figure 7.** Indoor localization models.

In this study, we are assuming that two or more light sources emit signals (ideally at least four). These light sources are visible and distinguishable when captured with a smart phone. The transmitter (Tx) shall transmit Manchester encoded data at a minimum frequency of 1 kHz to avoid direct or indirect flicker. According to IEEE 802.15.7, the on and off switching should be performed at a minimum rate of 200 Hz to avoid harmful effects on the eyes. When a smartphone passes through an emitter, the transmitter's signal is projected onto the camera. Although the frequency of the source is far above the frame rate of the camera, the signals can still be decoded due to the rolling shutter effect. CMOS sensors [46] that use a rolling shutter mechanism expose one or more columns at a time and scan one column at a time. When an OOK-amplified light source shines on the camera, distinct stripes of bright and dark appear in the image. The width of the bars depends on the scan time and, significantly, on the transmission frequency. We use an image processing stream, as described in Section 4.2.4. LEDs decoding algorithms to determine the transmitter, and the region of interest of each bulb (RoI) to decode the incoming data in each continuous frame were photographed. The data of each light source will be aggregated into final data representing a location to be located.

### 4.1. Transmitter Section

4.1.1. Transmitter Design

We built a transmitter system consisting of hardware devices commercial LED luminaires including LED ceiling lights, LED panel lights, and ESP32—WROVER microcontroller, as shown in Figures 8 and 9.

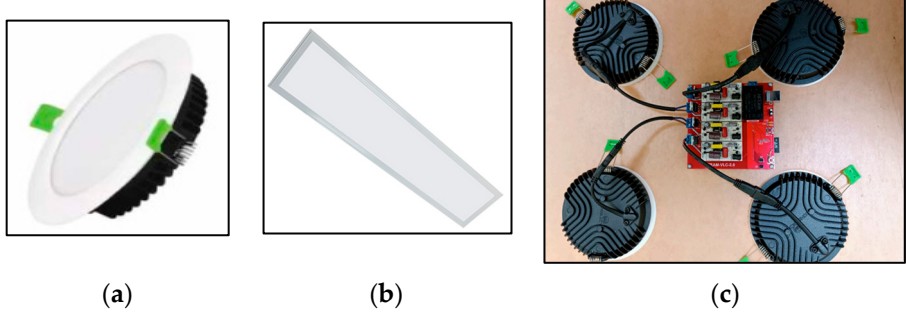

| (**a**) | (**b**) | (**c**) |

**Figure 8.** (**a**) LED downlights are utilized to locate users in the room; (**b**) LED panels are utilized to locate users in the hallway; (**c**) driver for LED lights as transmission.

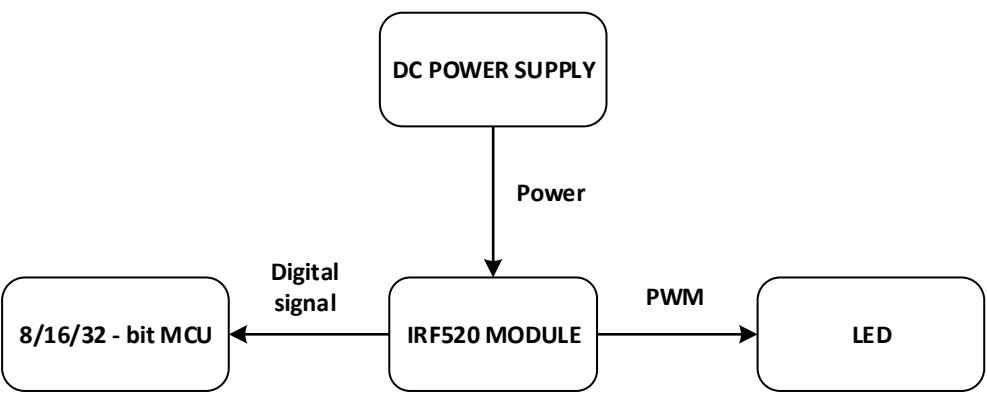

**Figure 9.** Diagram principle of Tx.

4.1.2. Transmitter Light-Emitting Diode Modulation

In visible light communication (VLC) modulation studies, on-off keying (OOK) modulation is one of the most popular techniques to provide data communication due to its ease of implementation. Our research focuses on indoor positioning with the method that each set of LED downlights or each LED panel will carry an ID to determine the location of the LED. Therefore, each sent message consists of a series of logical bits, 0 and 1. The demodulation method needs to decode the 0 and 1 bits from the bright/dark striped image by image processing. In this modulation technique, logic bits, 1 and 0, are converted into on- and off-duty cycles. Pulses in OOK modulation of a VLC system represent on and off states of light. Thus, an OOK modulation can be achieved simply by converting the received data into on-off states of light. However, OOK modulation still exhibits several drawbacks, particularly concerning flickering when representing extended data strings with numerous '0' and '1' bits, thereby limiting data transmission speed. Flickering issues can be tackled by increasing higher frequencies of on-off cycles and using a suitable bit encoding technique like Manchester encoding. In this modulation method, each '0' and '1' bit is represented by the sequence of symbols '01' and '10', respectively. This design choice ensures that there are no more than two consecutive matching symbols, effectively eliminating the flickering associated with OOK modulation. Additionally, the balanced representation of '0' and '1' bits remain consistent regardless of whether the data have been encrypted. Figure 10 illustrates the distinction between standalone OOK modulation and its combination with Manchester encoding. In the former case, a prolonged sequence of '0'

bits may produce noticeable flickering if the modulation frequency is insufficiently high, while in the latter case, the symbol ratio is more evenly distributed.

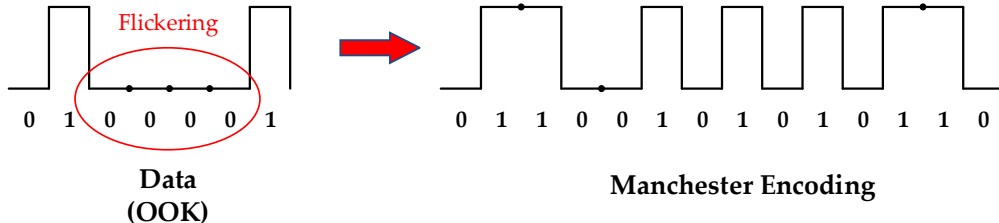

**Figure 10.** Eliminate harmful flickering by utilizing Manchester code.

In OOK modulation, LEDs are switched on and off according to the bits in the data string. This signal generation method relies on the rolling shutter mechanism of the CMOS sensor. The content and the number of signal bits in a data packet are determined according to the specific decoding method employed. However, the fundamental principle of these data packets remains consistent: they are transmitted and repeated an appropriate number of times to prevent data loss. The length of each packet is constrained to ensure that each LED image captures at least one complete packet. This packet structure includes a preamble and payload, as shown in Figure 11. Each data packet has a fixed number of bits for transmission, including 6 bits for the preamble (which is always fixed) and $n$ bits for the payload (utilizing Manchester encoding). The decision to consistently encode 6 preamble bits has two primary purposes: establishing frequency and distance parameters. In OOK modulation for VLC systems using the camera for the receiver, varying distances between the LED and the camera result in changes to the size of the LED in the image, impacting modulation performance, as shown in Figure 12.

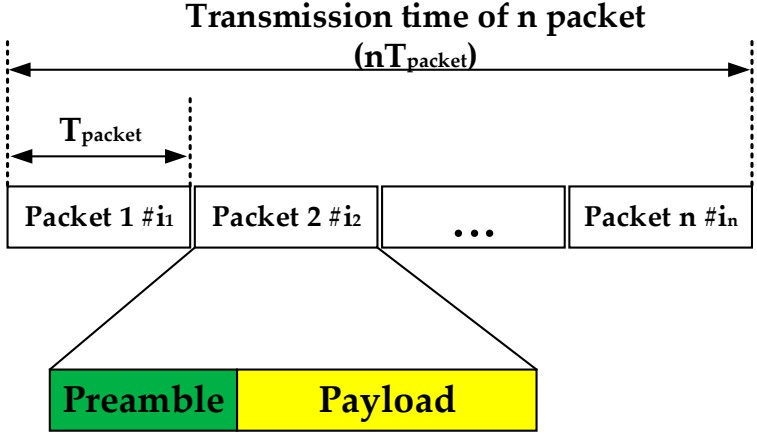

**Figure 11.** The structure of a data packet.

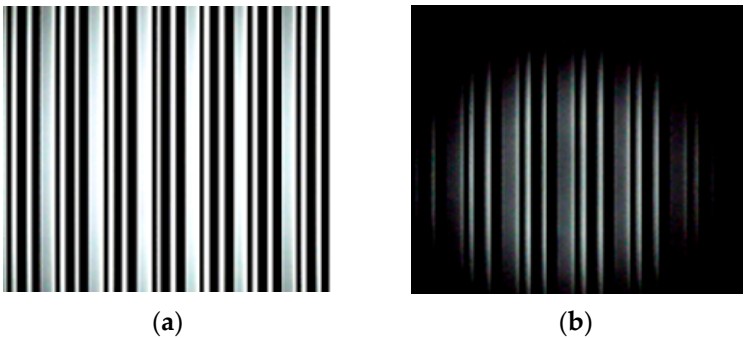

**Figure 12.** LED image at distances: (**a**) 50 cm and (**b**) 200 cm.

Additionally, the LED flashing frequency must be at least 200 Hz. However, experimental evidence indicates that a minimum frequency of 800 Hz is required to ensure the presence of at least one packet in the LED image at a distance of 50 cm, without causing a flicker perceptible to the human eye. This underscores the importance of ensuring that the maximum number of consecutive "0" bits does not exceed 3.

Moreover, the choice of a 6-bit preamble aligns with the IEEE 802.15.7—2018 standard governing preamble fields for PHY I, PHY II, and PHY III layers. Achieving full packet demodulation necessitates maintaining a sufficiently short distance between the camera and the LED unit. The maximum achievable distance is defined as the point at which the camera receives data equivalent to the number of packets in a single image. During this process, the camera continuously captures LED images and applies an image processing algorithm to detect the preamble 6 bits. Subsequently, it demodulates the amount of data required for a single packet. For a comprehensive understanding of the LED transmission data modulation scheme, refer to Figure 13. Initially, a string of 2 or 4 bits carrying the ID code for transmission is generated. After applying the Manchester encoding technique, the string's length doubles to 4 bits (corresponding to the original 2 bits) or 8 bits (corresponding to the original 4 bits). Before signal transmission, 6 preamble bits are added to the string to create a complete sequence of 10 bits or 14 bits. At this point, the ESP 32 module functions as a driver for the LED light, generating a PWM pulse based on the pre-established bit sequence. The pulse frequency can be adjusted through timer/counter interruptions, allowing the LED to emit a signal following the modulated bit sequence.

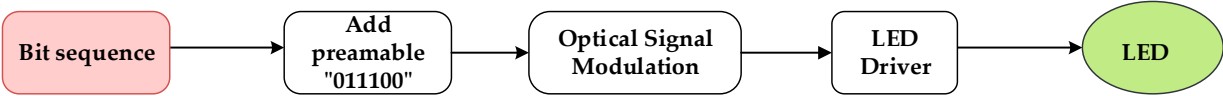

**Figure 13.** Modulation diagram of Tx.

*4.2. Receiver Section*

4.2.1. Smartphone CMOS Image Sensor's Rolling Shutter Operation

The working mode of the CMOS image sensor requires exposure time and data reading is carried out by scanning the pixel row by row, as shown in Figure 14. By switching between the on and off states of the LED during data transmission, bright and dark stripes will appear in the frame captured by the CMOS image sensor. The color of these stripes represents received bits. Dark stripes are for logic 0, and bright stripes are for logic 1. The width of dark and bright stripes will be equal; however, for Manchester encoded bits, there could be a maximum of two consecutive 0 s or two 1 s, which will have twice the width of one 0 or one 1 bit. For example, if the width of dark and bright stripes is 10 pixels individually, then the Manchester encoded bit pattern '011001' will have widths like 10, 20, 20, and 10 pixels.

In addition, the readout time value is the time it takes to read data from each row of pixels. To read data from each row of pixels, the corresponding voltage signal is converted to a digital signal by an analog-to-digital converter (ADC). The time required to read data from each row of pixels depends on the conversion rate of the ADC and the number of pixels in the sensor.

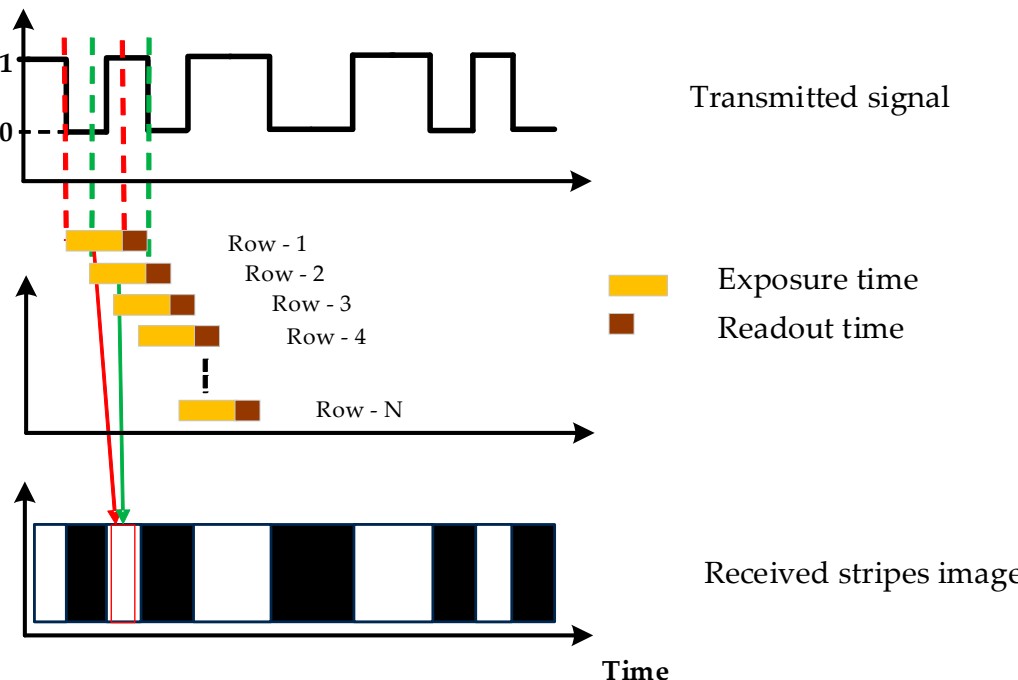

**Figure 14.** Rolling shutter operation of smartphone CMOS image sensor.

4.2.2. Impact of Camera Parameters

- Frame rate

The frame rate of the camera plays an important role in visible light communication (VLC), as it has a direct effect on the data transfer rate in the VLC system [47]:

$$R_{bps} = R_{fps} \times N_{bpf} \tag{1}$$

where $R_{bps}$ is the bit rate (bits per second), $R_{fps}$ is the frame rate of the camera, and $N_{bpf}$ is the number of bits per frame.

- Shutter speed

The shutter speed of the camera on a smartphone uses an electronic shutter. It works by turning the sensor on and off during exposure time. Shutter speed affects the pixel Signal-to-Noise Ratio (SNR) of the VLC system [47]. The longer the exposure time, the more light is exposed to the sensor, resulting in a brighter image as well as a higher pixel SNR, as shown in Figure 15. However, for images captured of motion objects, then increasing the exposure time will cause the image to be intentionally blurred to give a sense of movement. For a high-frequency flicker LED source in VLC systems, the captured image needs to have a distinct separation of bright/dark stripes. This led to the choice of reducing the exposure time to freeze movement, causing that movement to be captured as stillness in the image.

- Rolling rate

In smartphone cameras, rolling speed is a very important parameter and it is represented by the readout time value. This value to compute the thickness of a bit (1 or 0) determines how many pixels correspond to the smallest width of a bright/dark stripe in the data transmission technique in this study and it is not provided in any commercial product. Hence, to compute the thickness of a bit (1 or 0), we can use the following equation [47]:

$$N_{pixel/bit} = \frac{T_{clock}}{T_{rolling}} \tag{2}$$

where $N_{pixel/bit}$ is the thickness of 1 bit, $T_{clock}$ is the transmit cycle, and $T_{rolling}$ is the sensor's reading cycle. Due to Nyquist's theorem, the condition to transmit high-speed data using the rolling shutter effect is $T_{clock} \geq 2 \cdot T_{rolling}$.

- Focal length

Focal length is the distance between the point of convergence of your lens and the sensor or film recording the image. It has a direct impact on the size of the resulting image produced by the LED, as shown in Figure 16. The relationship between focal length and distance is shown as follows:

$$\frac{d[m]}{H_{LED}[m]} = \frac{f[mm]}{h_{LED[mm]}} \tag{3}$$

where f is the camera focal length, d is the transmission distance, $H_{LED}$ is the actual LED size, and $h_{LED}$ is the LED size in the captured image.

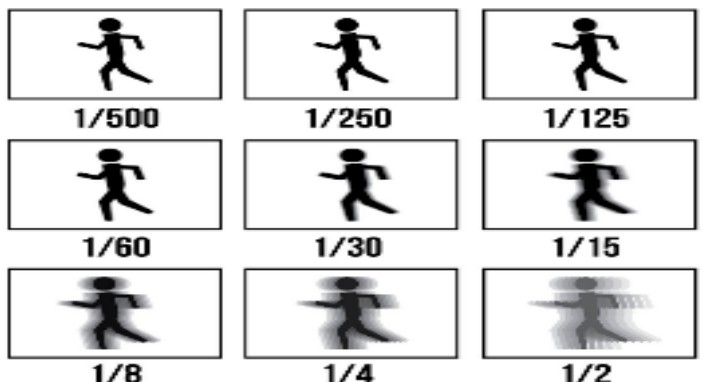

**Figure 15.** Effect of exposure time on motion.

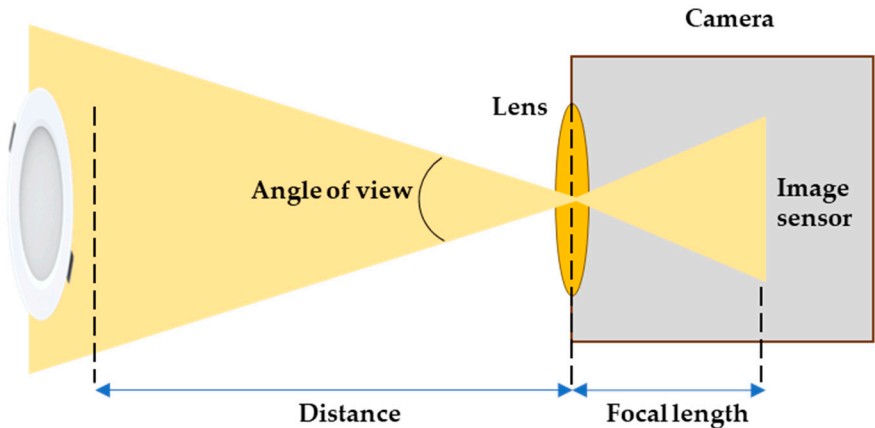

**Figure 16.** Focal length diagram.

### 4.2.3. Smartphone Receiver

In our VLC system, we have developed a camera application for the receiver on the Android Studio platform. We have automated the configuration of several camera parameters to capture images with the lowest Signal-to-Noise-Ratio (SNR). When configuring the camera, we mainly focus on two parameters: exposure time and autofocus. Table 2 presents the parameters of the components utilized in the receiver section.

**Table 2.** Receiver component parameters.

| Parameter Name | Values |
|---|---|
| Smartphone Model | Google Pixel 4 |
| CPU | $1 \times 2.84$ GHz and $3 \times 2.42$ GHz and $4 \times 1.78$ GHz (Qualcomm Snapdragon 855) |
| GPU | 257 MHz |
| Image Sensor | Rolling Shutter CMOS |
| Frame Rate | 30 fps |
| Camera | Front camera: 8 megapixels Back camera: 16 megapixels |
| Focal length | 28 mm—f/1.7 48 mm—f/2.4 |
| Aperture | f/1.7; f/2.4 |
| Camera API | Camera2 API level 30 |
| Camera Image Resolution | $1080 \times 2220$ px |

We emphasize that camera configurations will be supported by software, not hardware. Two key parameters, exposure time and autofocus, will be configured using the Camera2 API. It is provided by Google for Android devices, enabling interaction with and control over camera features. Upon opening the application, we configured the automatic light balance mode to be turned off. This is to reduce the exposure time of the image sensor to a minimum, ensuring a clear separation between the illuminated area of the LED and the non-light source areas, minimizing noise in the decoding process, as shown in Figure 17. In addition, the autofocus mode enables fast autofocus on each image.

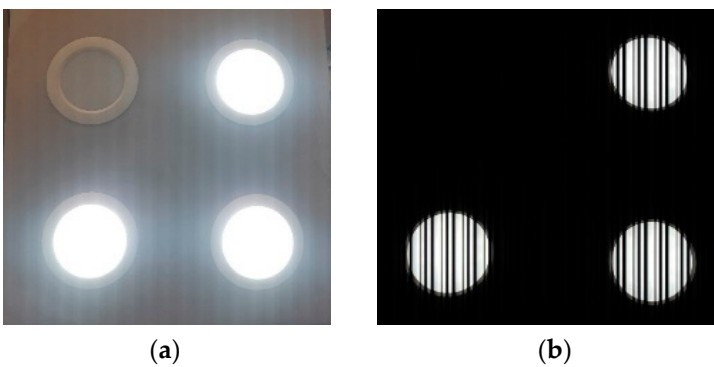

(**a**)　　　　　　　　　　(**b**)

**Figure 17.** Difference with automatic exposure modes: (**a**) on and (**b**) off.

4.2.4. Light-Emitting Diodes Decoding Algorithms

Our receiver system (Rx) is software for smart mobile devices on the Android operating system and is called LEDs-ID with the 1.2.0 version. This name is based on the goal of determining the ID for each LED cluster. The software takes advantage of the camera integrated with a CMOS sensor to capture images of light sources that carry data continuously. Subsequently, we employ computer vision techniques directly on the mobile phones to decode data from each captured frame. Presently, the speed of image processing is reliant on two key components: the CPU and GPU. We leverage the inherent hardware resources of the smartphone for local image processing, harnessing the device's processing power without requiring external server support. However, when the system is tasked with complex processing operations, the frame rate may experience a reduction. This reduction is primarily due to the necessity of handling intricate graphics computations. As a result, the frame rate may decrease by around 25–33%, as shown in Figure 18.

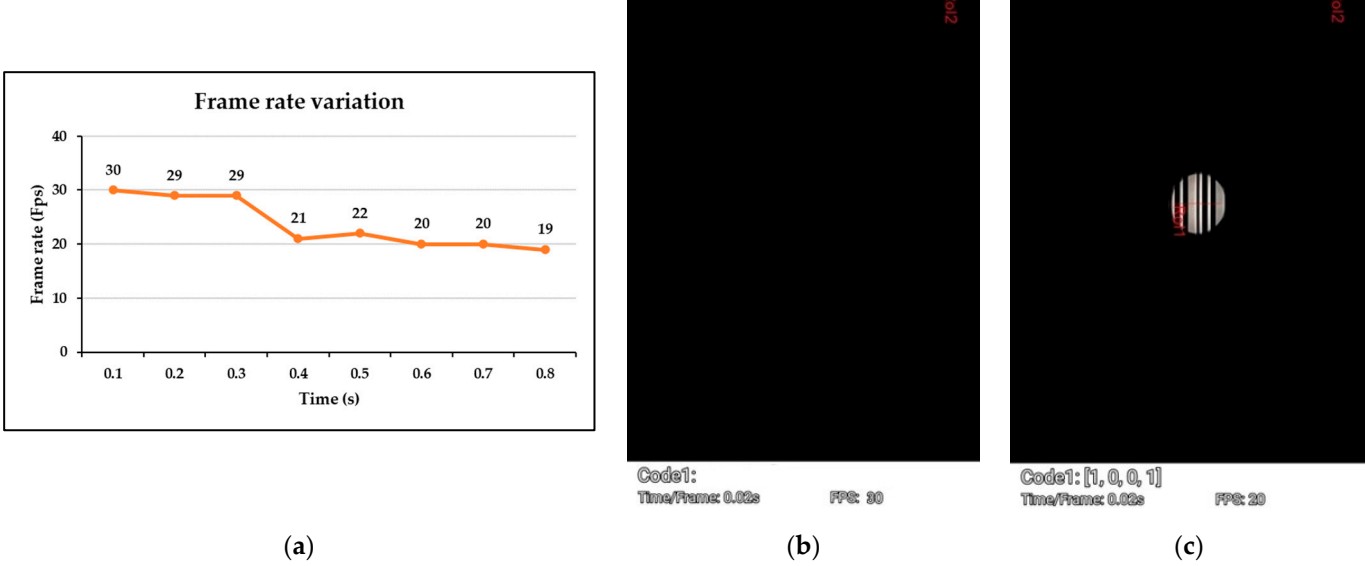

**Figure 18.** (**a**) Measure the change in frame rate of your Google Pixel 4 phone before and after applying image processing, (**b**) FPS is 30 when not image processing, (**c**) FPS is 20 when image processing.

By analyzing the image captured through the camera integrated with the CMOS sensor, it becomes evident that the primary objective of the receiver is to decode the transmitted message frame, symbolized by alternating bright and dark stripes. The overall processing steps outlined in Figure 19 provide a systematic approach to decode the data from each individual image. As shown in the figure, the employment of the image processing method mandates the execution of nine distinct steps, which collectively culminate in the successful completion of the decoding process.

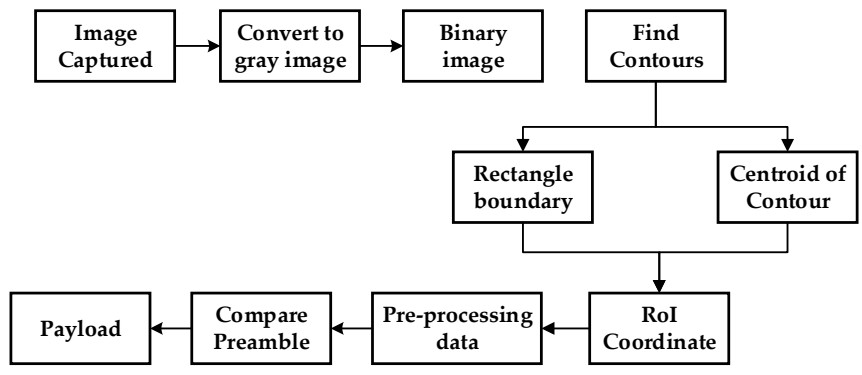

**Figure 19.** Block diagram of LED decoder.

The original image captured by the smartphone CMOS image sensor is first converted to a grayscale image, as shown in Figure 20b. We applied different methods from the step of converting from gray images to binary images. In this process, before converting to binary images, we use the image blur technique by convolving the image with a low-pass filter kernel to reduce noise. The size of the kernel we chose is 5 × 5 which is relatively small because the decoded objects are bright/dark stripes with very small widths (only a few pixels); therefore, using this size will still preserve many details of the image. In the algorithm to find the image thresholding, several algorithms are utilized and evaluated for effectivity such as manual thresholding, adaptive threshold, binary threshold, Otsu thresholding, and combined binary threshold and Otsu. The experiments point out that they do not give the desired results. To evaluate the attributes and values of all pixels on a gray image, we experiment with the average value of all gray pixels while eliminating

pixels with very small values starting from value 0. Since, when calculating the threshold value by averaging all gray pixels, the number of pixels with a value of 0 is always 10–12 times larger, leading to the final results having an increased error from 10% to 15%. The optimal value chosen is 10 with experiments ranging from 1 to 20, providing the best final results as shown in the article. The grayscale image is further processed into a binary image using a method that utilizes the mean threshold with $T_{threshold} = \frac{\sum_{j=0}^{N} I_j}{N}$, where $T_{threshold}$ is the threshold that needs, $I_j$ is the intensity of each pixel in a gray image, and N is the number of pixels with a luminous intensity greater than 10. Pixels with a gray level above the threshold are set to 255, whilst the rest are set to 0 as shown in Figure 20c. This produces a white object on a black background. Starting from the binary image, the contour finding method is applied to detect white striped objects in Figure 20d. Once contours are identified, the algorithm determines the rectangular contour for each object and computes the center of each contour in Figure 20f,g. A rectangle is then drawn around each contour to create precise and efficient bounding boxes for the objects within the image. This process yields accurate coordinates for both the top and bottom edges of the rectangle surrounding each object. These two values will have two main tasks: determining the width of the smallest bright stripe ($N_{pixel/bit}$) and determining the region of interest (RoI) of the LED in Figure 20h. The width of a bright stripe is calculated in pixels. We emphasize that this is a key value in decoding the data on each image captured when using this decoding algorithm because the bright stripes are the objects representing logical bit 1. In fact, the $N_{pixel/bit}$ value on different smartphones may be different because it depends on the rolling rate of the camera. Therefore, determining a method to calculate $N_{pixel/bit}$ that is compatible across all smartphone ROM versions is also a challenge. In [48], Nam et al. proposed a method to calculate $N_{pixel/bit}$ by measuring the coordinates of the starting point and ending point of RoI and counting the number of bright/dark stripes the RoI line passes through. From there will be the length of the RoI line and the number of stripes. Then, Nam et al. applied the following formula:

$$N_{pixel/bit} = \frac{d_{stripes}}{n_{stripes}} \tag{4}$$

where $d_{stripes}$ is the length of the RoI line, $n_{stripes}$ is the number of stripes on the RoI line.

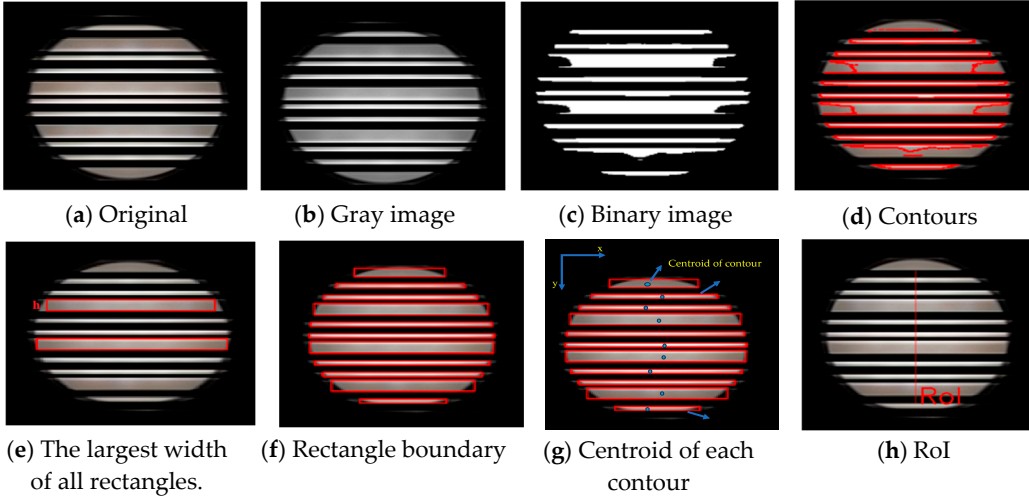

(**a**) Original    (**b**) Gray image    (**c**) Binary image    (**d**) Contours

(**e**) The largest width of all rectangles.    (**f**) Rectangle boundary    (**g**) Centroid of each contour    (**h**) RoI

**Figure 20.** Image processing pipeline. From the original image captured to determine the region of interest (RoI).

Nam et al.'s proposal highlights the need for an alternative solution capable of automatically computing $N_{pixel/bit}$ to accommodate mobility when applied in indoor positioning systems. In our proposed packet content structure, the bright stripe with the largest width

is configured to carry three logic 1 bits which are in the preamble of the proposed packet content. As shown in Figure 20e, red rectangles represent the widest bright stripes containing three logic 1 bits. Furthermore, we know the coordinates of the top and bottom edges of the rectangle that surrounds the bright stripe. Hence, we propose the following formula to calculate $N_{pixel/bit}$:

$$N_{pixel/bit} = \frac{h_{max}}{3} \tag{5}$$

where $h_{max}$ is the largest width of all rectangles. In addition, the coordinates of the centroid of each contour play an important role. When combined with the coordinates of the upper and lower edges of the enclosing rectangle, they will determine the regions of interest (RoI) on an image to be processed. Computing the coordinates of the centroid of each contour will be shown in detail as follows:

$$M_{ij} = \sum_i \sum_j x^i y^j I(x, y) \tag{6}$$

From formula (6), $M_{00}$ is determined:

$$M_{00} = \sum_i \sum_j I(x, y) \tag{7}$$

where $M_{00}$ is the number of pixels other than 0. The centroid of each contour coordinates is as follows:

$$\{\bar{x}, \bar{y}\} = \left\{ \frac{M_{10}}{M_{00}}, \frac{M_{01}}{M_{00}} \right\} \tag{8}$$

where $M_{10}$ and $M_{01}$ are the first-order moments.

Due to the image as the light source of the LED, we define the region of interest (RoI) as a straight line passing through the area with the highest brightness that meets the specified condition in Figure 20h. As a result, crucial values for defining the RoI line are the coordinates of the starting and ending points. After obtaining the coordinates of both the starting and ending points of the RoI line, the RoI line's outcomes under different scenarios are illustrated in Figure 21a,b. When processing images involving an LED cluster of two or more lights, we encountered the challenge of simultaneously handling multiple discrete light sources. This necessitated the establishment of distinct RoI lines for each individual LED. As depicted in Figure 21c,d, certain situations involve an RoI line passing through two highlights. To solve this, we measured the distance and angle of deviation between the RoI line's initial and terminal coordinates. With experiments from 60 cm to 200 cm and frequency from 1 to 10 kHz, the number of bits received by an LED downlight with data transmission per frame is 10–15 bits. In addition, the diameter of the LED is 10 cm and the design of the LED downlight cluster has a distance between the two closest lights of 35 cm, showing that this distance is always larger than the size of an LED on the collected image, and the minimum number of bits on an RoI line passing through two LEDs is 30 bits. Therefore, the condition of the distance between the starting and ending points of the RoI line for each light is as follows: $D < 30 \times N_{pixel/bit}$. Subsequently, we distinguish and sketch independent RoI lines for each LED, as shown in Figure 21e–g. This approach enabled us to effectively manage the presence of multiple light sources.

We will process the RoI line as it passes through the pixel values within the grayscale image. This processing will yield an array of pixel values for decoding, which in turn must be converted back into binary format. To achieve this, we plan to utilize a highly optimized thresholding algorithm, enhancing the accuracy of the resulting binary data. The algorithm of the thresholding scheme is the third-order polynomial curve fitting. Assume each element in the array of grayscale pixel values on the RoI line is $(x_i, y_i)$, where $x_i$ is the index of the pixel value in the array, $y_i$ is the grayscale value of that pixel, $i = 1, 2, 3, \ldots, n$

(n is the length of the pixel array). The third-order polynomial fitting curve $f(x_i)$ is shown in Equation (9):

$$f(x_i; a_0; a_1; a_2; a_3) = a_0 + a_1 x_i + a_2 x_i^2 + a_3 x_i^3 \qquad (9)$$

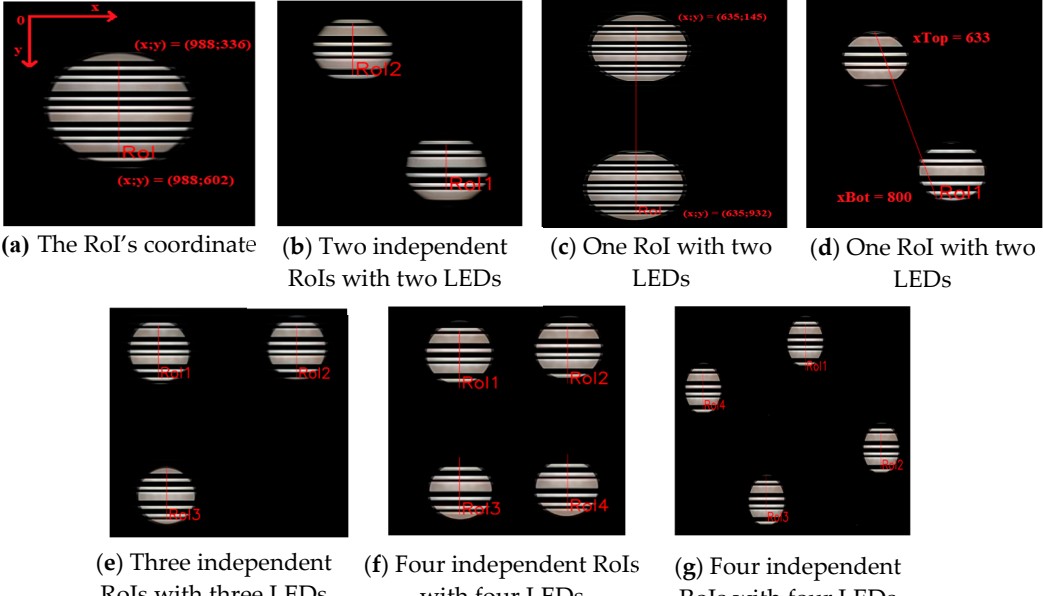

**(a)** The RoI's coordinate

**(b)** Two independent RoIs with two LEDs

**(c)** One RoI with two LEDs

**(d)** One RoI with two LEDs

**(e)** Three independent RoIs with three LEDs

**(f)** Four independent RoIs with four LEDs

**(g)** Four independent RoIs with four LEDs

**Figure 21.** Result of the RoI with the LED cluster, after and before the establishment of distinct RoI lines.

Then, the square deviation is shown in Equation (10):

$$[y_i - f(x_i)]^2 \qquad (10)$$

And the total square deviation E can be represented in Equation (11):

$$E(a_0; a_1; a_2; a_3) = \sum_{i=1}^{n} [y_i - f(x_i)]^2 = \sum_{i=1}^{n} \left[ y_i - \left( a_0 + a_1 x_i + a_2 x_i^2 + a_3 x_i^3 \right) \right]^2 \qquad (11)$$

By setting $\frac{\partial E}{\partial a_0}, \frac{\partial E}{\partial a_1}, \frac{\partial E}{\partial a_1}, \frac{\partial E}{\partial a_2} = 0$, we can obtain four simultaneous equations; hence, we can solve these equations and obtain the values of $a_0; a_1; a_2; a_3$. After finding the values of $a_0; a_1; a_2; a_3$ in Equation (9), a third-order polynomial curve can be constructed as shown in Figure 22a. This curve is considered to be the threshold corresponding to every pixel on the RoI line and is calculated for the different light sources. This processing does not affect the total execution time very much and is in the range of 20–25% execution time. Whereas, when it is processed by a multi-thread, it is more time-consuming because the CPU spends more time allocated for the thread stack, for the setup of thread-specific data structures, and synchronization with the operating system. At each pixel position, if the grayscale value is above the third-order polynomial curve (threshold), logic 1 is recorded; if the grayscale value is below the curve, logic 0 is recorded.

After applying curve fitting, we obtain an array of binary values. This array will contain the data sent from the transmitter, which includes a 6-bit preamble and either a 4-bit or 8-bit payload, as shown in Figure 22b. The ultimate decoding task involves examining every consecutive set of 6 bits within the array against the 6-bit preamble. If a complete match is found, the corresponding payload will be successfully retrieved in Figure 22c.

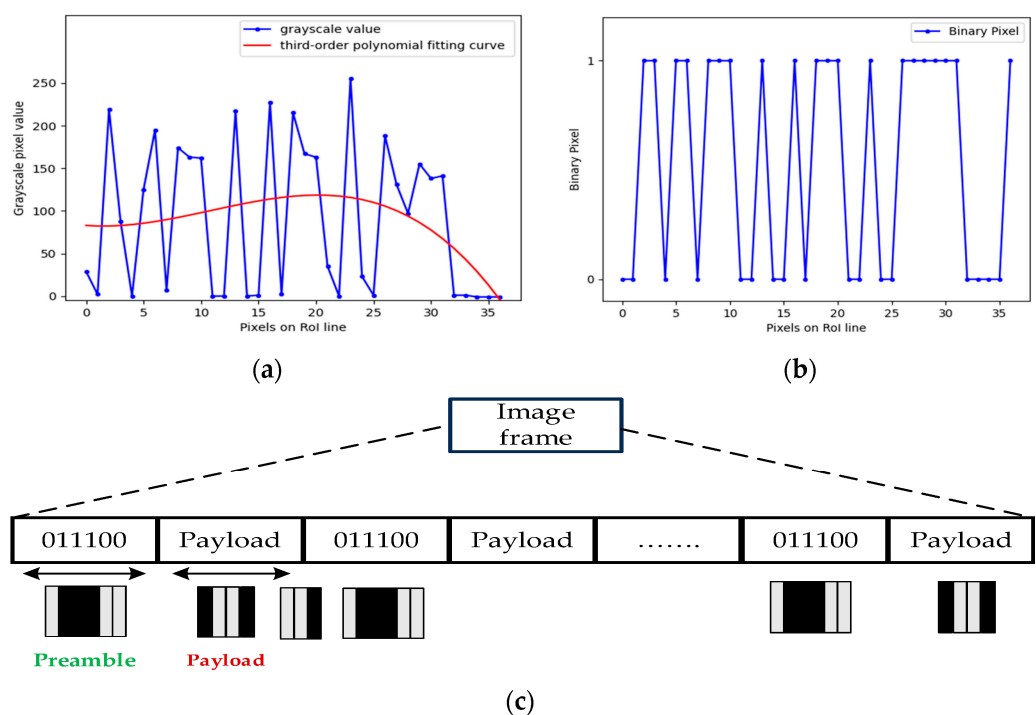

(a)

(b)

(c)

**Figure 22.** (**a**) 3rd order threshold, (**b**) array of 0-bit and 1-bit values, (**c**) image frame showing bright and dark fringes representing the packet preamble and payload is included.

## 5. Experiment and Results

### 5.1. Experimental Setup

We propose to test the system in the experimental work area with the transmitter being downlight LEDs cluster and panel LEDs, while the receiver uses a smartphone (Google Pixel 4) in Figure 23.

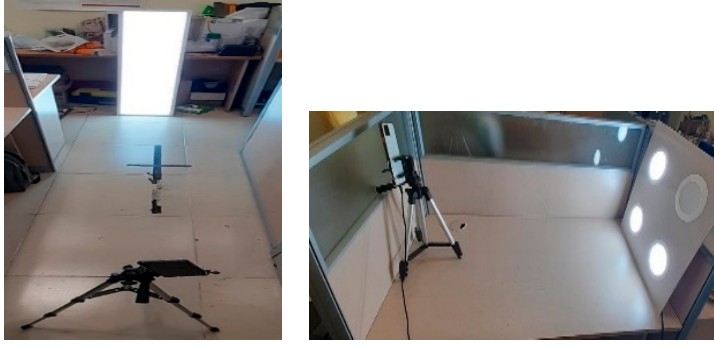

**Figure 23.** Experiential test field with Tx is the LED and Rx is the smartphone.

In this test, the parallel LEDs in the LED cluster need to be designed with a suitable distance to achieve the highest transmission accuracy rate with a ceiling distance from 300 cm to 400 cm. In Ref. [48], Nam et al. proposed that the center distance of the two closest LEDs from 35 cm to 50 cm as shown in Figure 24 will achieve the highest transmission speed with distances from 200 cm to 300 cm. To achieve such results, Nam et al. conducted an experiment in which Tx is a cluster of two LED downlights and Rx is a webcam connected to software programmed in the LabVIEW environment.

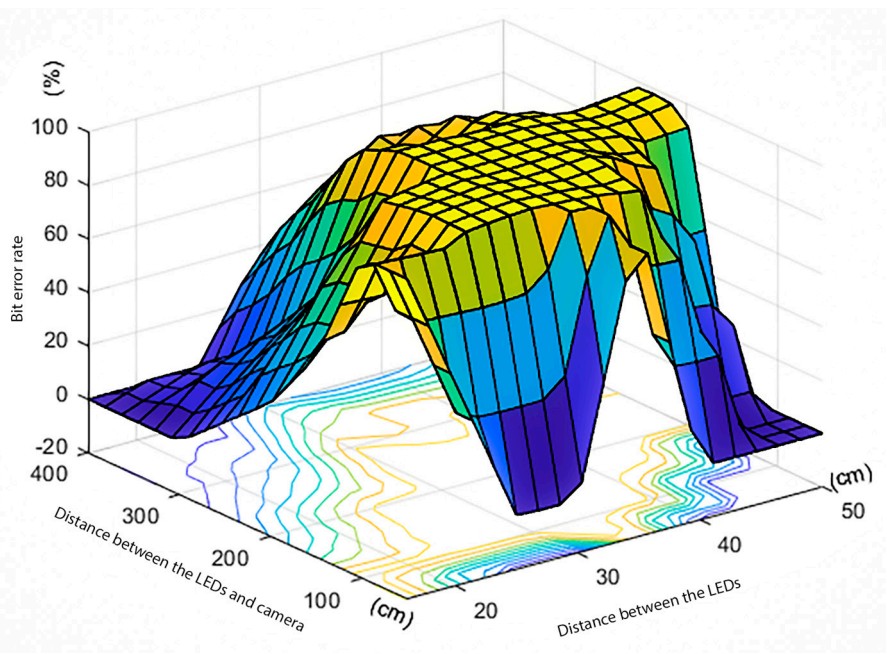

**Figure 24.** Bit error probability of two LEDs and distance between two parallel LEDs.

The results obtained from Nam et al. are very important in designing LED clusters for indoor positioning applications. Based on this result, we design LED light clusters with two or more LED lights with a fixed distance of 35 cm between two closest LEDs for lighting and data transmission, as shown in Figure 25. Then, we test the bit error rate in data transmission and reception at each distance between the LEDs or LED panel and the smartphone in the range from 100 cm to 210 cm. In addition, as analyzed in the previous sections, the LED flicker frequency will affect the transmission and reception distance with a low error rate. Therefore, experimenting with different LED flicker frequencies will be necessary to determine the most optimal frequency for data transmission and reception in this system.

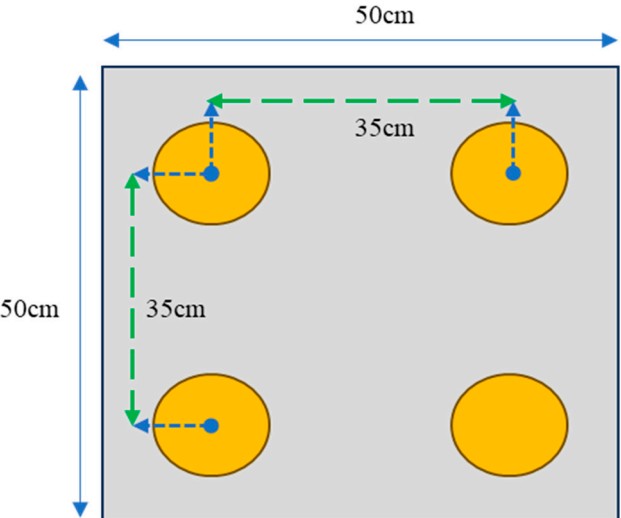

**Figure 25.** Design of the LED cluster.

*5.2. Experimental Results*

First, we measure the bit rate when there is only one LED transmitting data with two scenarios as follows: Scenario 1.1 will have an LED light source acting as the data transmission source. In scenario 1.2, in addition to a single LED light source transmitting

data, there is an additional light source that is always bright as a noise source, as shown in Figure 26. The LED transmits data in a sequence of 10-bit at 8000 Hz, which includes 6-bit preamble and 4-bit payload.

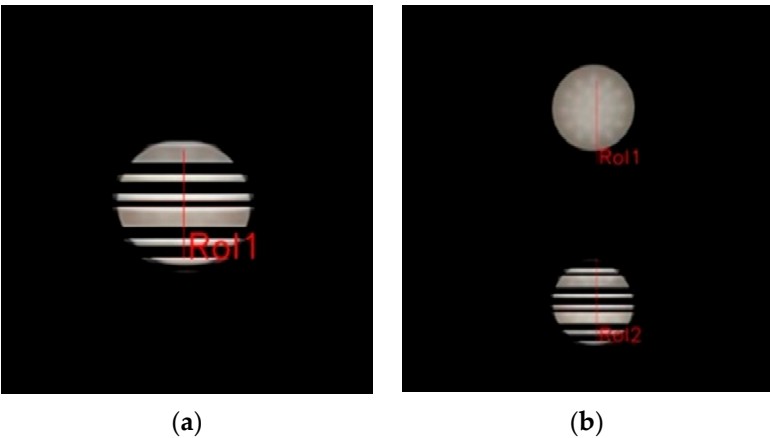

(**a**)                                        (**b**)

**Figure 26.** The scenario transmitting with one LEDs: (**a**) one LED is transmitting; (**b**) one LED is transmitting and one LED is a noise source.

BER in the two scenarios shown in Figure 27 exhibits a linear relationship with distance during transceiving. Within the range from 100 cm to 190 cm, the BER remains relatively low, fluctuating between 1% and 16% in both cases. However, as the distance exceeds 210 cm, the BER increases proportionally. This is attributed to the increased distance causing the LED image to shrink, resulting in a reduced number of bright (dark) stripes. Consequently, the reception rate for each transmitted frame (10 bits/frame) decreases. While the presence of noisy light can influence the bit error rate, the most significant divergence between the two experiments is only $\Delta = 4\%$.

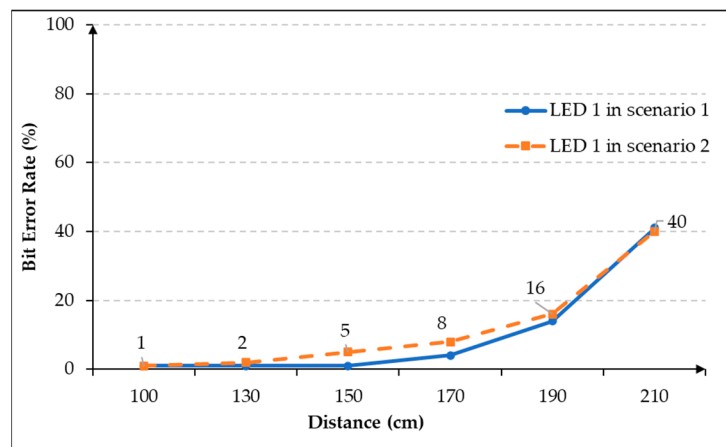

**Figure 27.** Bit error probability in the two scenarios with a single transmitting LED.

With an LED cluster with two or more data transmission light sources, we need to measure and evaluate the bit error rate in different situations in terms of the angle between the cluster and the smartphone camera in Figure 28. Here, we are positioning the smartphone's camera parallel to the LED light source and adjusting the angle of deviation between the smartphone and the LED light within that parallel plane.

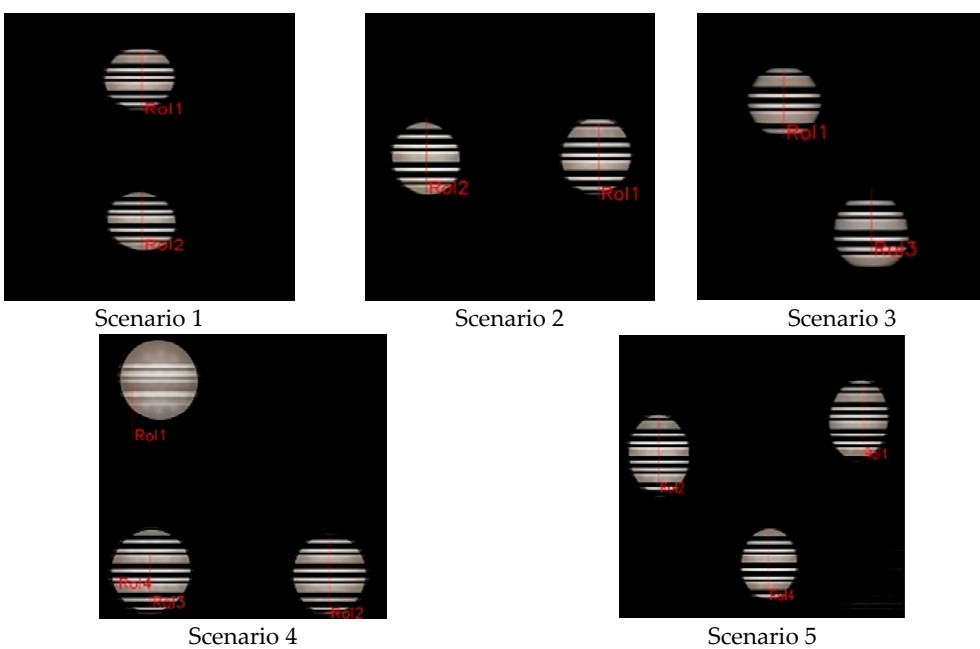

**Figure 28.** Image of five scenarios with two or more transmitting LEDs.

Data in scenarios with two or more LEDs transmitting data will be provided in 10-bit sequences with four different payload bits for each light and at the same frequency of 8000 Hz. In addition, in scenario 4, we continue to test and measure the bit error rate when there is one noisy light. In Figure 29, the line chart compares bit error rates between transmission and reception scenarios. In general, the bit error rate is relatively low in the range of 1–20% at distances from 100 to 190 cm. For scenario 4, when there are additional lights causing interference, the bit error rate of each light increases very slightly by about 2–3% in different locations. Therefore, using two or more data transmission lights in an environment with normally bright lights will have little effect on data transmission quality.

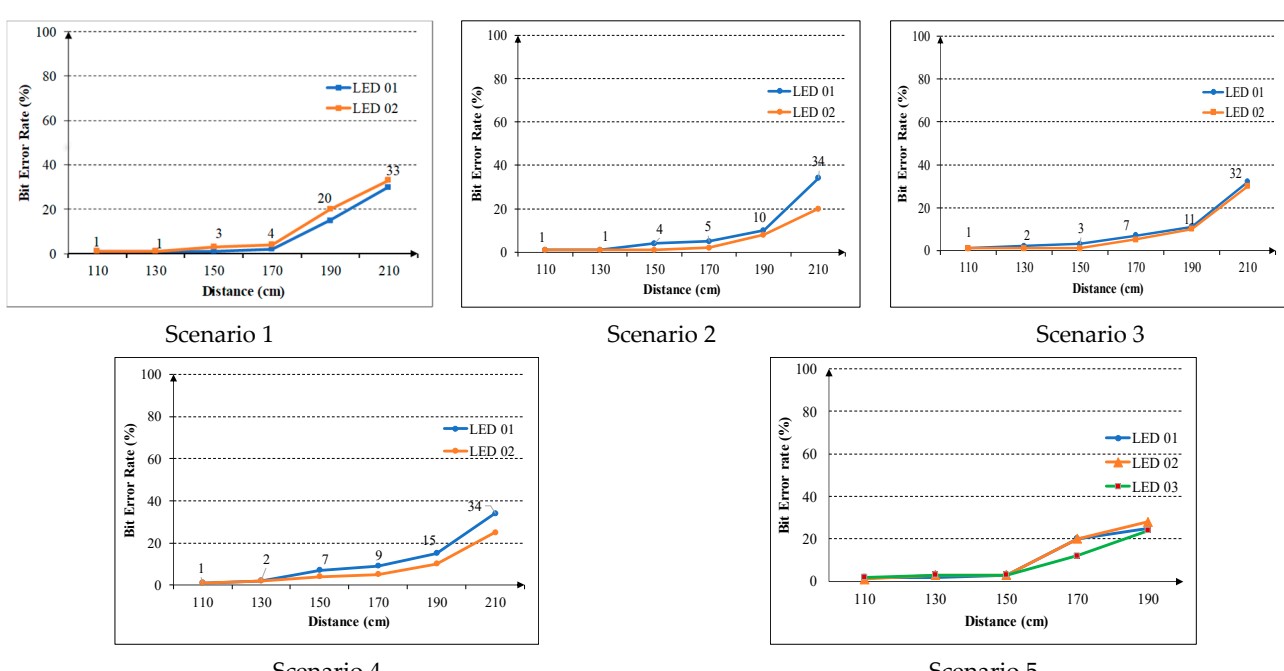

**Figure 29.** Bit error probabilities in five scenarios with two or more transmitting LEDs.

For indoor systems that use LEDs as a light source, in addition to downlight LEDs, panel LEDs are very commonly used. In our experiments with panel LED lights, we test two distinct scenarios, both involving data transmission at a consistent frequency of 8000 Hz. In scenario 1, the LED continued to transmit data using a 10-bit sequence, with a payload consisting of 4 bits in Figure 30a. However, in scenario 2, we altered the data transmission to employ a 14-bit sequence, featuring an 8-bit payload in Figure 30b. Our decision to increase the data transmission parameters in scenario 2 was driven by the unique characteristics of panel LED lights. These lights tend to capture a greater number of bright and dark stripes in their imagery compared to downlight LED. Hence, we deemed it feasible to take advantage of this increased capability for data transmission.

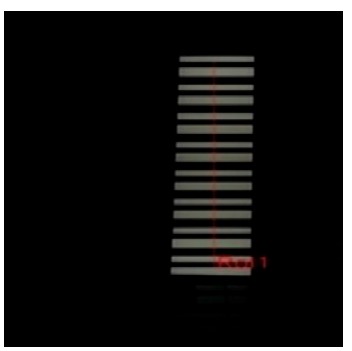 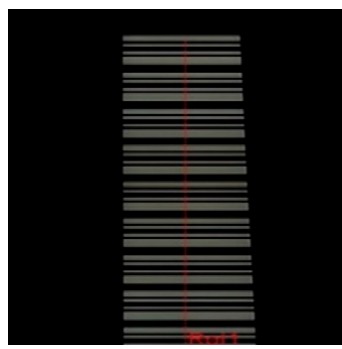

(**a**) Transmit data with 10-bits sequence.     (**b**) Transmit data with 14-bits sequence.

**Figure 30.** Image panel LED with two transmitting scenarios.

When experimenting with LED panel lights, the trend of the graph in Figure 31 is similar to the previous graphs but we see a significant increase in the 1–20% bit error rate distances. For scenario 1, the bit error rate is only 1–2% at 230 cm. This is the distance used when designing the lighting system, to achieve the greatest accuracy when transmitting at 2–2.5 m, due to the ceiling height in the range of 3–4 m. In addition, with 8-bit data transmission experiments, the bit error rate increases by about 10% with the same distance. The reason for the increase is that the number of black and white stripes has significantly increased, namely, the array of 0 and 1 values after using curve fitting are equivalence points with values very close to the curve fitting, which are the values that affect the curve fitting; thus, increasing BER.

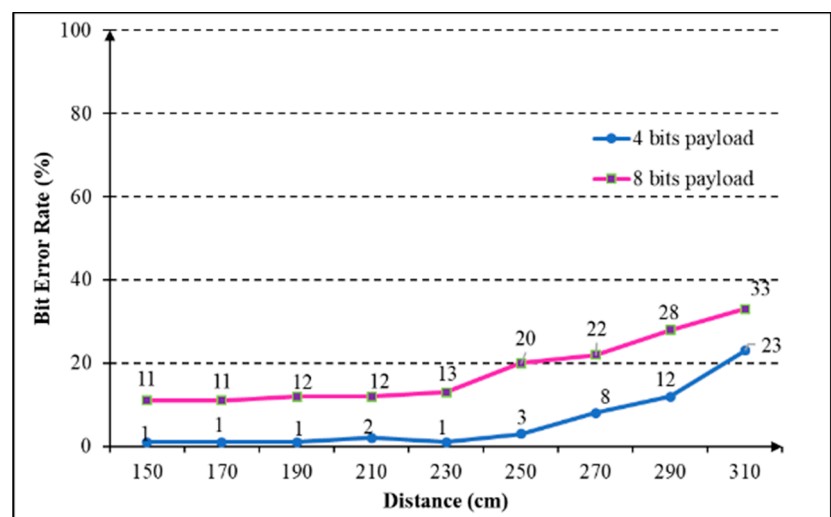

**Figure 31.** Bit error probability in the two scenarios with the transmitting LED panel.

In contrast to previous scenarios, in Figure 32, we utilize a surface plot to illustrate the data transmission rate for LED downlights. The color scale transitions from green

to dark blue, representing an accuracy transceiver rate ranging from 0% to 100%. This visualization is the result of experiments conducted at varying distances between the smartphone and the LED light source, while also adjusting the LED flashing frequency. The figure demonstrates that as the flashing frequency of the LED increases to its maximum value, the decoding success rate also increases with distance. At flicker frequencies ranging from 2000 to 4000 Hz, achieving a high accuracy rate of over 80% requires a relatively short distance of only 30–80 cm. This phenomenon can be because as the distance increases, the lower LED flicker frequency results in fewer bright/dark stripes appearing on each frame. This reduced stripe count leaves insufficient information for successful decoding. The number of bright stripes remains constant since the width of each stripe does not change with a fixed flicker frequency. However, the size of the LED image diminishes with increased distance. In addition, when the flicker frequency exceeds 10 kHz, there is a downside to the reduced width of the bright stripes, which leads to fewer pixels on one bright stripe. Consequently, as the distance continues to increase, the accuracy of $N_{pixel/bit}$ calculation and decoding decrease significantly. At distances of 190–210 cm, even though there are numerous bright/dark stripes present in the image, the decoding accuracy rate may drop to about 40–60%

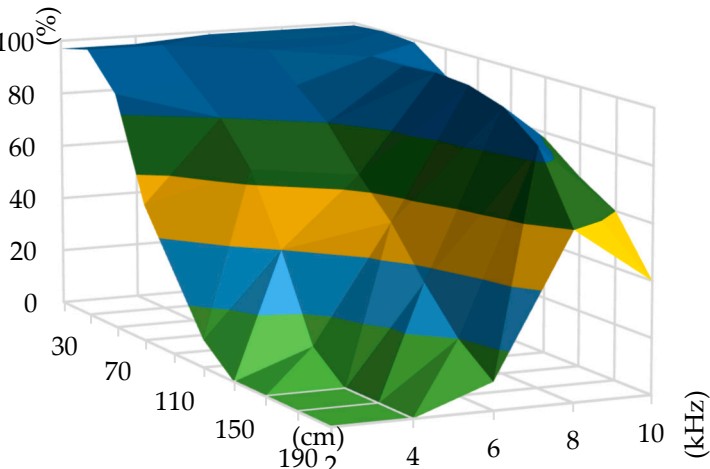

**Figure 32.** The successful data transmission rate of LED downlights with different frequencies.

To achieve the highest transmission accuracy within a distance range of approximately 150–250 cm, an LED flicker frequency of approximately 8000 Hz proves to be the optimal choice. This frequency ensures that the $N_{pixel/bit}$ value is sufficiently high, enabling effective data decoding even when the distance is extended to about 150–200 cm. Thus, in indoor positioning applications, a flicker frequency of around 8000 Hz emerges as the most advantageous selection for transmitter design.

## 6. Discussion and Conclusions

In this study, we implemented a system to decode data transmitted from LEDs in a VLC system that relies on a smartphone's CMOS image sensor and a slightly modified LED illumination system, which supports indoor positioning optimizing OOK modulation. Through experiments, we established the relationship between distance, data transmission speed, the number of image lights, and the quality of communication, all encapsulated in concise data containing unique IDs. The results of our system demonstrate the feasibility of applying this method in smart home applications. However, we acknowledge certain limitations in our system. Firstly, the decryption software is not compatible with all ROM versions using the Android operating system, which restricts its scalability for practical use. The software is designed as a receiver using cameras to capture images and decode them, aiming for deployment on the different Android smartphone versions. However, with high commercialization demands, the vendors often customize software from Google's original

Android version and have differences in hardware design. Therefore, obtaining camera parameter values for calculation and changing to suit research applications will be difficult on many versions. A general method for performing this task will not be compatible with all Android versions after customizing. Secondly, the decoding process has not optimally addressed the issue of noise from various light sources. Despite these drawbacks, none of them appear to be insurmountable. Overall, our system has illustrated the practicality of employing this fundamental method. Future work will focus on determining the smartphone terminal deflection angle and orientation relative to the LED, combined with the decoded ID, to obtain the final user coordinates.

**Author Contributions:** Conceptualization, Q.D.N. and N.H.N.; data curation, Q.D.N.; formal analysis, Q.D.N. and N.H.N.; funding acquisition, N.H.N.; project administration, N.H.N.; software, Q.D.N.; supervision, N.H.N.; validation, Q.D.N.; writing—original draft, Q.D.N.; writing—review and editing, Q.D.N. and N.H.N. All authors have read and agreed to the published version of the manuscript.

**Funding:** This research was funded by Hanoi University of Science and Technology (HUST) under the project number TC2022-PC-011.

**Institutional Review Board Statement:** Not applicable.

**Informed Consent Statement:** Not applicable.

**Data Availability Statement:** Data are contained within the article.

**Conflicts of Interest:** The authors declare no conflict of interest.

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
