# Peer review of "Mobile Application for Visible Light Communication Systems: An Approach for Indoor Positioning"

_photonics, doi:10.3390/photonics11040293_

Round 1

Reviewer 1 Report

Comments and Suggestions for Authors
  • Why Manchester RRL? Why not a more efficient codification?
  • How is the driver implemented? How is power transferred to the LED during modulation?
  • shown above. Where? image reference. - line 510
  • Figure 18. Why did you choose these steps instead of fewer steps or different methods?
  • Figure 17 suggests a comparison before and after applying image processing, but no comparison is seen in the figure.
  • N is the number of pixels with a luminous intensity greater than 10. Why 10? - line 570
  • We established specific conditions for each LED to distinguish and sketch independent RoI lines. What conditions? - line 619
  • The fitting curve is calculated each time for the different light sources? Is it not a very time-consuming recognition?
  • line 681 - both figure names are wrong
  • Figure 28. The explanation does not specify whether the smartphone is being rotated in pitch, roll, or yaw (although it can be inferred from the images). The angles used to capture each image are also missing.
  • It is mentioned to have payloads of 4 or 8 bits, but these are not effective as the Manchester encoding was used. Thus, the payloads have 2 or 4 effective bits, which allows the assignment of four to sixteen different IDs. Is this sufficient for final applications?
  • The flicker frequency of about 8000 Hz turns out to be the most advantageous choice for transmitter design. Is this conclusion scalable to any light source at the same distance?
  • The decryption software is not compatible with all ROM versions using the Android operating system. What is the reason?
Comments on the Quality of English Language
  • line 30 - Commu-nication (typo)
  • line 86 - Comparasion (typo)
  • line 527 - app-lication (typo)
  • line 571 - ablove (typo)
  • line 583 - I In fact (typo)
  • This software continuously captures images using a light source. - line 545
    This text is confusing, are you using a flashlight to capture the images?

Reviewer 2 Report

Comments and Suggestions for Authors

The authors have designed an indoor wireless communication system by employing LEDs to emit the information (in the visible spectrum) and smartphones to decode it.

The work is interesting, however, it lacks an emphasis on improvement brought in by the research on existing communication systems. I believe if the following comments are answered, it might bring some improvement to the article.

1) Very few references are provided for the work's historical background. Please add more references (preferably from journal articles).

2) The literature review for the work is also limited.

3) Some figures can be combined to reduce the number of figures, such as Figure and Figure 2, Figure 4 and Figure 5, Figure 8 and Figure 9, etc.

4) Figures are not crisp (especially Fig. 7, Fig. 28). Increasing pixel size would be a better approach. Moreover, use a larger font so that the image can be read properly.

5) Authors say that longer exposure time results in clear images but Fig. 14 shows that an exposure time of 1/2 results in blurred images whereas an exposure time of 1/500 results in clear image.

6) The definition of focal length is not correct. Authors say "Focal length is the distance from the image sensor to the lens in an optical system.", however, the focal length is the distance from the center of the lens to the image when parallel rays enter the lens. The position of image sensor can be changed but the focal length of the lens cannot be changed (f=-R/2).

7) Explain a bit more about the limitation of the camera system when it is only going to 30 fps. Why should we use the proposed methodology when we can only capture information at 30 fps? The authors should also mention that how much data is decoded when 8000 Hz flicker is used.

8) In Fig. 28 and Fig. 30, only one letter seems to be transmitted as evident from the pattern. There is no variation.

9) If data is transmitted at 8000 Hz, please state how much actual data is transferred in one second when the preamble is separated from the payload. Are the results any better than the data transfer rate of RF communication?

10) Conclusions do not emphasize any key findings numerically.

11) Axes are not fully mentioned in Figure 32.

12) Why is there a different bit error probability in Figure 29? Are the distances between LEDs and the sensor consistent?

Comments on the Quality of English Language

The following comments on English should be addressed.

1) There should be no dash (-) in communication as written on Line 30.

2) In Line 171, "... but only one fixed can..." only one fixed what?

3) In Line 527, the application is written as "app-lication".

4) I think there is some error in Line 659 "In Nam et al. proposed the center...".

5) More such errors should be checked throughout the paper.

Reviewer 3 Report

Comments and Suggestions for Authors

in one line: The paper analyzes different approaches to indoor positioning technologies and propose an indoor positioning system that applies the advantages of VLC technology. Based on my overall appreciation, there are some issues which require to be addressed further in order to improve the quality of the manuscript. 1. Chapter 3 reviews the literature with a general focus on the following topics: Modulation techniques in VLC systems and comparing indoor positioning technology using VLC systems with RF based indoor positioning systems. However, a detailed review of the manuscript's contribution is not apparent. I recommend that the authors clarify the problem addressed and justify it with recent articles. 2. Figure 7 should be improved. I recommend representing the location model with block diagrams that provide more information to the reader. 3. What is the difference between the methods used in this paper (modulation, decoding, etc.) and the methods reported in the literature? explain in the manuscript 4. slightly modified LED illumination system, which supports indoor positioning with higher accuracy than previous research efforts. 5. In the conclusions section, the authors make the following statement"..., slightly modified LED illumination system, which supports indoor positioning with higher accuracy than previous research efforts" however, the results do not have figures where the positioning performance is compared with previous work. 6. Improve the quality of the figures and add names to the axes. Figures 24 and 32. 7. Review the structure of some bibliographic citations: References 5, 6, 18.

Round 2

Reviewer 2 Report

Comments and Suggestions for Authors

The axes of Figure 32 should have titles, for example, "(cm)" is written on one axis but which length is it representing? Please correct this image.